# “What Did I Tell This Sad Person?”: Memory for Emotional Destinations in Korsakoff’s Syndrome

**DOI:** 10.3390/jcm12051919

**Published:** 2023-02-28

**Authors:** Mohamad El Haj, André Ndobo, Ahmed A. Moustafa, Philippe Allain

**Affiliations:** 1Laboratoire de Psychologie des Pays de la Loire (LPPL-EA 4638), Faculté de Psychologie, Nantes Université, Chemin la Censive du Tertre—BP 81227, CEDEX 3, 44312 Nantes, France; 2Unité de Gériatrie, Centre Hospitalier de Tourcoing, 59200 Tourcoing, France; 3Institut Universitaire de France, 75000 Paris, France; 4Marcs Institute for Brain and Behaviour, School of Psychology, University of Western Sydney, Penrith, NSW 2751, Australia; 5Department of Human Anatomy and Physiology, The Faculty of Health Sciences, University of Johannesburg, Johannesburg 2092, South Africa; 6Laboratoire de Psychologie des Pays de la Loire (LPPL EA 4638), SFR Confluences, Maison de la Recherche Germaine Tillion, Université d’Angers, 5 bis Boulevard Lavoisier, CEDEX 01, 49045 Angers, France

**Keywords:** Korsakoff’s syndrome, destination memory, emotion, memory

## Abstract

We investigated destination memory, defined as the ability to remember to whom a piece of information was previously transmitted, for emotional destinations (i.e., a happy or sad person) in Korsakoff’s syndrome (KS). We asked patients with KS and control participants to tell facts to neutral, positive, or negative faces. On a subsequent recognition task, participants had to decide to whom they told each fact. Compared with control participants, patients with KS demonstrated lower recognition of neutral, emotionally positive, and emotionally negative destinations. Patients with KS demonstrated lower recognition of emotionally negative than for emotionally positive or neutral destinations, but there were no significant differences between recognition of neutral and emotionally positive destinations. Our study demonstrates a compromised ability to process negative destinations in KS. Our study highlights the relationship between memory decline and impaired emotional processing in KS.

## 1. Introduction

Korsakoff’s syndrome (KS) is the chronic phase of acute Wernicke disease as caused by thiamine deficiency, malnutrition, and prolonged heavy ingestion of alcohol [1,2]. KS is mainly associated with a severe decline in episodic memory and, to some extent, a decline in emotional processing. However, one open question is whether the declines in episodic memory and emotional processing are related in KS. The current study addresses this challenge by investigating the relationship between the decline of destination memory and decline of emotional processing in KS. To this aim, we first introduce the notion of destination memory as well as the effects of emotion on this memory system. We then highlight research demonstrating how KS affects destination memory and the processing of emotional, especially negative, information. We hypothesized that patients with KS would demonstrate difficulty in remembering negative emotional destinations.

Destination memory refers to the ability to remember to whom a piece of information was previously transmitted (e.g., “did I send that email to John or Mary?”) [3,4,5,6]. The concept of destination memory was made prominent by the pioneering work of Koriat and colleagues [7,8], who investigated the tendency of older adults to tell the same story over and over. These researchers attributed this tendency to a weak integration of information with its encoding context. Supporting this assumption, research on destination memory tends to incorporate this ability into contextual memory, as destination recall requires attributing information to its appropriate context, that is, the receiver [9,10]. The link between destination and contextual memory has been empirically demonstrated by studies attributing a decline of destination memory, as observed in several neurological and psychiatric populations, to a decline in the ability to bind information from several contexts (e.g., special or temporal) into a coherent episode [11,12,13,14,15]. In these studies, as well as in other research in general, destination memory has been evaluated by asking participants to tell facts to pictures of celebrity faces and then, on a subsequent recognition task, to identify to whom they had told each fact.

Destination memory can be influenced by emotion, and research has demonstrated that the ability to remember to whom a piece of information was previously told can be influenced by the emotional characteristics of the destination. One study investigated this issue by asking heathy young and older adults to tell neutral facts to a neutral, an emotionally positive, and an emotionally negative face [16]. Afterward, participants were invited to remember the facts that were previously told to each face. Older adults demonstrated higher destination memory for facts told to negative than to positive faces as well as higher destination memory for facts told to positive than to neutral faces. In contrast, younger adults demonstrated similar memory for neutral and both types of emotional destinations. In other words, unlike younger adults, older adults tended to place higher emphasis on emotional faces relative to neutral faces, exhibiting higher memory for emotional than for neutral destinations. Using similar procedures, one study demonstrated the beneficial effects of emotion on destination memory in healthy older adults but not in patients with Alzheimer’s disease [17]. The lack of an effect of emotion on patients’ destination memory was attributed to a decline in function of the amygdala, a brain region involved in processing emotional information [18,19], which was reported to be impaired in Alzheimer’s disease [20]. Building on this research, we investigated whether patients with KS would demonstrate difficulty remembering emotional destinations.

Besides being based on research demonstrating how destination memory can be influenced by emotion, the current study is based on research demonstrating the negative effects of KS on both destination memory and emotion. The effects of KS on destination memory were investigated in a study in which patients with KS and control participants were invited to tell proverbs to pictures of celebrities, and in a subsequent recognition test, they were invited to indicate to which celebrity they had previously told the proverbs [21]. The results demonstrated lower destination memory in patients with KS than in control participants, which is related to a decline of episodic memory in KS. Relatedly, KS is mainly associated with severe impairment of episodic memory [1,2,22,23,24]. In addition to episodic memory decline, patients with KS tend to display impaired emotional and affective functioning. For instance, patients with KS tend to display impaired social inference ability [25] as well as some socio-affective symptoms, such as apathy, disinterest, loss of initiative, and decreased social desirability [2]. Furthermore, patients with KS tend to demonstrate difficulties categorizing stimuli (e.g., pictures and words) according to their affective valence (i.e., neutral, positive, and negative) [26] as well as difficulties decoding negative voices, such as those depicting anger and fear [27]. Critically, patients with KS tend to demonstrate difficulty recognizing emotional expressions in the faces of others.

The impairment in recognition of facial emotional expressions in KS was investigated by Oscar-Berman et al. [28], who investigated the ability of patients with KS to identify the emotional expressions of faces expressing neutral or one of several emotional expressions (i.e., happy, sad, or angry). The results demonstrated difficulty recognizing facial expressions in patients with KS; further, patients with KS attributed more emotional intensity to facial expressions compared to control participants. Similar difficulties were reported in a study in which patients with KS were invited to recognize the emotions of faces depicting neutral or one of several emotional expressions (i.e., anger, disgust, fear, happiness, sadness, and surprise) [29]. Patients with KS demonstrated an impairment of facial expression recognition, with higher difficulties for negative emotions. These findings suggest difficulty in patients with KS with processing negative facial expressions [30]. This difficulty may result in an impairment to processing negative destinations, that is, a difficulty to remember whether information was previously told to a negative destination.

To summarize, we investigated whether emotion would influence destination memory in patients with KS. Although previous research has demonstrated a decline in destination memory in patients with KS [21], they were solely invited to process neutral destinations. We therefore investigated whether destination memory in KS may be influenced by emotional characteristics of destination. This issue is important for several reasons. Theoretically, the relationship between the decline in memory in general and impaired emotional processing in KS is an open question, as research tends to investigate either memory decline or emotional/affective disorders in KS. At a social level, destination memory is intimately linked with social cognition [3,4,5,6]. In our daily lives, we constantly relay information to colleagues, friends, family members, and/or strangers. The decline in the ability to remember to whom specific information was previously told (i.e., decline in destination memory) negatively affects communicative efficacy and daily interactions with others [4]. The study of destination memory in KS may thus contribute to the understanding of social interactions in patients with KS, especially when they have to attribute information to its appropriate (e.g., happy or sad) interlocutor. To this aim, we invited patients with KS and control participants to tell facts to a neutral, an emotionally positive, and an emotionally negative face. Afterward, we invited participants to remember the facts they previously told to each face. We expected lower destination memory in patients with KS than in control participants; we expected that these difficulties would be specifically observed for emotionally negative destinations, which was based on research demonstrating difficulty processing negative information in KS [27].

## 2. Method

### Participants

The study included twenty-five patients diagnosed with KS (sixteen women and nine men; M age = 57.04 years, SD = 5.49; M years of formal education = 9.82, SD = 3.42) who were recruited from several alcohol-dependence or psychiatric medical units and day-care facilities at Lille and Nantes. As for the control group, we recruited 28 participants without previous or current substance addiction and without psychiatric or neurological history (17 women and 11 men; M age = 55.18, years, SD = 4.66; M years of formal education = 9.11, SD = 3.20). The control group was matched with the KS group according to sex ratio (χ^2^(1, *N* = 53) = 0.61, *p* > 0.10), age (*t*(51) = 1.35, *p* > 0.10), and educational level (*t*(51) = 0.72, *p* > 0.10). The diagnosis of KS was made by experienced psychiatrists using the DSM V [31] criteria for alcohol-induced persisting amnestic disorder. The diagnosis was based on an extensive history of alcoholism and nutritional depletion, notably thiamine deficiency. All patients with KS were in a chronic (more than one-year post-onset) and stable condition but had no confounding Wernicke psychosis at the time of testing or signs of alcohol-related dementia [32]. The patients underwent a cognitive examination, which is described in Section 3. This cognitive examination served to confirm amnestic syndrome in the KS patients as well as normal cognitive performance in the control participants. All participants provided informed consent. Ethical approval was obtained from the national French ethical committee (approval number: 2022-A02482-41).

## 3. Procedures

### 3.1. Cognitive Evaluation

As shown in Table 1, we evaluated episodic memory, working memory, and verbal fluency. We evaluated episodic memory with the task of Grober and Buschke [33] in which participants were invited to retain 16 words, each of which describes an item (e.g., piano) that belongs to a different semantic category (e.g., musical instrument). Afterward, as a distraction condition, participants were invited to count backwards from 374 in 20 s. Counting was followed by two minutes of free recall; participants were invited to recall, as much as possible, the 16 words, and the score from this phase provided a measure of verbal episodic memory. As for working memory, we used the digit span task [34], in which participants were invited to repeat a string of single digits in the same order (i.e., forward spans) or in reverse order (i.e., backward spans). As for verbal fluency, participants were allocated one minute to generate as many words as they could beginning with the letter “P”. Proper nouns and variations on words (e.g., “psychologist” and “psychology”) were not allowed. The score was the number of correctly generated words.

### 3.2. Destination Memory

We replicated procedures used in research on destination memory in general [35,36,37,38] and, more specifically, in research on the effects of emotion on destination memory [16,17].

## 4. Materials

We used twenty-six facts (e.g., Rome is the capital of Italy) and twenty-six high quality pictures. The pictures contained unfamiliar faces and were taken from the FACES database [39]. Among the twenty-six faces, eight depicted happiness, eight depicted sadness, and ten depicted neutral facial expressions (eight + two for the training task). Note that the emotional valences of the faces was assessed by Ebner, Riediger and Lindenberger [39], who controlled the facial expressions of the faces regarding the positions of the muscles around the eyes, nose, and mouth. Because women tend to demonstrate better processing of female faces than male faces, and probably vice-versa for men [40], half of the faces were male, and the other half were female. Because older adults tend to encounter more difficulties when processing younger faces than older faces, and vice-versa for younger adults [41,42,43,44,45,46], and because this own-age bias was also reported for destination memory [47], the ages of the faces were matched to those of the patients with KS (*p* > 0.1) and those of the control participants (*p* > 0.1). With these precautions taken, the performance of our participants was likely to reflect the processing of the emotional valences of faces rather than issues of gender or age bias. Concerning facts, they were judged as neutral (M = 0.33, SD = 0.66) by a separate sample of eight middle-aged adults (four women and four men, M age = 56.90 years, SD = 6.44) on a five-point scale (−2 = negative, 0 = neutral, +2 = positive). Each fact was printed in black Times New Roman 48-point font under one (16 × 16 cm) colored face on white paper of A4 format.

## 5. Assessment

The assessment of destination memory included three phases: study, distraction, and recognition. The study phase included 24 trials. In each trial, the experimenter presented a fact that participants were invited to tell the paired face. Immediately after the study phase, the participants were engaged in the distraction phase, which consisted of reading strings of three-digit numbers aloud for one min. Immediately after the distraction phase, they proceeded to the recognition phase, in which the previously-exposed 24 face–fact pairs (12 intact/original and 12 re-paired/new) were presented, one at a time. For each pair, participants were invited to decide, with no time constraint, whether they had previously told the particular fact in the pair to the associated face or not. Encoding was intentional, and participants were instructed that their memory for the facts and destinations would be tested later. In order to ensure that participants could comfortably execute the task, we invited them to perform a training trial prior to the experiment, during which they were asked to tell two facts to two faces to subsequently decide whether they previously told the particular fact in the pair to the associated face or not. As recommended for analyzing recognition memory [48], performance referred to the proportion of hits (correct “yes” responses) minus the proportion of false alarms (incorrect “yes” responses). Hence, regardless of the nature of destination (neutral, positive, or negative), a score of 1 means that the participant recognized all of the pairs correctly without any false alarms.

## 6. Results

We investigated the differences between patients with KS and control participants regarding mean destination recognition as well as regarding recognition for each category of destination (i.e., neutral, emotionally positive, and emotionally negative destinations). We also investigated differences across the three categories of destination memory (i.e., neutral, emotionally positive, and emotionally negative destination) within each population. Because data were not distributed normally, as observed by Kolmogorov–Smirnov tests, the Mann–Whitney U test was used for intergroup comparisons, and Wilcoxon’s signed-rank test was used for intragroup comparisons. We provided effect sizes using Cohen’s *d* [49]: 0.20 = small; 0.50 = medium; 0.80 = large. Cohen’s *d* was calculated for nonparametric tests according to the recommendations of Rosenthal and DiMatteo [50] and Ellis [51]. For all tests, the level of significance was set as *p* ≤ 0.05; *p* values between 0.051 and 0.10 were considered to be trends.

### Low Memory for Destination Information in KS

Data is provided in Figure 1. Compared with control participants, patients with KS demonstrated lower mean destination recognition with means of 0.27 (SD = 0.47) and 0.81 (SD = 0.11), respectively (*Z* = −5.25, *p* < 0.001, Cohen’s *d* = 2.08). Compared to control participants, patients with KS also demonstrated lower recognition of neutral (*Z* = −2.09, *p* < 0.05, Cohen’s *d* = 0.60), emotionally positive (*Z* = −3.55, *p* < 0.001, Cohen’s *d* = 1.11), and emotionally negative destinations (*Z* = −5.34, *p* < 0.001, Cohen’s *d* = 2.15). Patients with KS demonstrated lower recognition of emotionally negative destinations than of emotionally positive (*Z* = −2.01, *p* < 0.05, Cohen’s *d* = 0.87) or neutral destinations (*Z* = −2.13, *p* < 0.05, Cohen’s *d* = 0.94), but we did not find any significant differences between recognition rates of neutral and emotionally positive destinations (*Z* = −0.26, *p* > 0.1, Cohen’s *d* = 0.10). Control participants demonstrated higher recognition of emotionally negative destinations than of emotionally positive (*Z* = −2.14, *p* < 0.05, Cohen’s *d* = 0.88) or neutral destinations (*Z* = −3.58, *p* < 0.001, Cohen’s *d* = 1.84) as well as higher recognition of emotionally positive than of neutral destinations (*Z* = −2.01, *p* < 0.05, Cohen’s *d* = 0.82).

## 7. Complementary Analysis

Considering hits vs false alarms, the analysis demonstrates lower hits in patients with KS compared to control participants with means of 0.47 (SD = 0.52) and 0.88 (SD = 0.02), respectively (*Z* = −4.23, *p* < 0.001, Cohen’s *d* = 1.90). KS patients showed higher false alarms compared to control participants with means of 0.20 (SD = 0.21) and 0.07 (SD = 0.02), respectively (*Z* = −3.41, *p* < 0.001, Cohen’s *d* = 1.10). In addition, to investigate potential gender differences, we compared the scores of women vs those of men for mean destination recognition; analysis demonstrated no gender differences between patients with KS (women: M = 0.24, SD = 0.43, men: M = 0.30, SD = 0.50, *Z* = −1.13, *p* = 0.26, Cohen’s *d* = 0.31) or between control participants and KS patients (women: M = 0.77, SD = 0.16, men: M = 0.85, SD = 0.17, *Z* = −1.32, *p* = 0.19, Cohen’s *d* = 0.37).

## 8. Discussion

We investigated the effects of emotion on destination memory in KS by asking patients with KS and control participants to remember whether they previously told information to a neutral, emotionally positive, or emotionally negative destination. Compared to the control participants, patients with KS demonstrated lower mean destination recognition as well as lower recognition of neutral, emotionally positive, and emotionally negative destinations. Patients with KS demonstrated lower recognition of emotionally negative than of emotionally positive or neutral destinations, but we did not find any significant differences between recognition of neutral and emotionally positive destinations. Control participants demonstrated higher recognition of emotionally negative than of emotionally positive or neutral destinations as well as higher recognition of emotionally positive destinations than of neutral destinations.

To begin with the low mean destination recognition in patients with KS, these findings mirror previous research demonstrating a decline in destination memory in KS, which was attributed to the compromised ability to integrate information with its encoding context [21]. There is a body of research demonstrating difficulties in patients with KS attributing information to its appropriate context, for instance, remembering the locations of drawings [52], the position of a spot of light presented on a board [53], the locations of pictures on cards [54], the locations of pictures of objects in grids [55], the locations of words presented in the corners of a screen [56], and the locations in which objects were placed [57]. These difficulties were also observed for autobiographical memory, that is, memory for personal information [58,59]. Patients with KS tend to demonstrate difficulties retrieving autobiographical memories situated in a specific spatiotemporal context [22,60,61,62,63]. These studies in the literature, and our findings, mirror the contextual account according to which the episodic memory decline in KS is caused by an inability to encode, store, or retrieve contextual information [1]. In addition to demonstrating low mean destination recognition, patients with KS in our study demonstrated low recognition of neutral, emotionally positive, and emotionally negative destinations. In other words, in this study, patients with KS did not benefit from the emotional characteristics of faces to improve their destination memory and reach the recognition level of control participants. Again, these findings can be attributed to contextual memory impairment, limiting the patients’ ability to use the emotional characteristics of faces as salient cues to improve their destination memory. Alternatively, the compromise of emotional processing in KS may also limit patients in this way.

In our study, patients with KS demonstrated lower recognition of emotionally negative than of emotionally positive or neutral destinations, but we did not find any significant differences between the recognition of neutral and emotionally positive destinations. These findings suggest a compromised ability to process negative information in KS. The difficulty to process negative information in KS was reported by research demonstrating a compromised ability of patients to recognize negative facial expressions [29] and to decode negative voices [27]. The decreased ability of patients with KS to process negative information may be a result of disruption to the amygdala, as KS has been associated with extensive impairments in limbic structures (i.e., amygdala and mammillary bodies) [2,64]. At the psychological level, the decreased ability of patients with KS to process negative information may be a result of an emotional regulation strategy by which patients avoid negative information to avoid the risk of exposure to negative effects. This assumption is supported by a study by Johnson et al. [65], who reported a preference for positive information over negative information in patients with KS. This being said, we do not pretend that patients with KS avoid negative information to inevitably seek a positive one, as our participants with KS demonstrated similar performance levels for neutral and emotionally positive destinations. The lack of superior memory for emotionally positive compared to neutral destinations in patients with KS can be attributed to the general disruption of affective information in KS, as the syndrome is generally characterized by apathy or a lack of initiative and/or interest in affective information [2]. Note, however, that patients with KS can experience some emotional experience upon retrieval, especially when retrieving emotional autobiographical memories [66,67,68].

The difficulties of patients with KS to process emotional, especially negative, information can be interpreted in light of the dual-process model [69,70], which proposes that adapted human behaviors are typically based on the interaction between two cerebral systems: (a) the “reflective system”, a controlled system underlying the cognitive processing of stimuli that relies on memory and executive functions and initiates controlled or deliberate decisions, and (b) the “affective–automatic system,” an appetitive system underlying the impulsive processing of information that triggers automatic responses based on associative learning (stimulus–response). The affective–automatic system encompasses an affective component involved in core affect decoding (e.g., recognition of facial expressions) and an automatic component involved in the attribution of aversive or pleasant values to environmental stimuli through conditioning. Based on this model, the difficulty of patients with KS to process negative emotionally destinations can be attributed to a decline in the reflexive system in the syndrome. More specifically, the reduced activation of this system may lead to lowered arousal levels in patients with KS when encountering negative information. However, this speculation should be confirmed by future research investigating the neural basis of destination memory in KS.

Unlike patients with KS, control participants demonstrated higher recognition of emotionally negative than of emotionally positive or neutral destinations as well as higher recognition of emotionally positive than of neutral destinations. The high memory for emotionally negative destinations in our control participants does not fit with the well-known positivity bias in healthy aging, although this bias may explain the higher destination memory of emotionally positive than of neutral destinations in these participants. Research using several methodological approaches supports the notion that older adults tend to prioritize emotionally positive compared to emotionally negative information [71,72]. This research leads to the assumption that older adults attend more to emotionally positive than emotionally negative information in order to maximize a positive mood state [73]. This assumption is rooted in the socioemotional selectivity theory, which posits that older adults react differently to positively emotional information because they are primarily motivated to pursue emotionally meaningful goals [74]. However, the positivity bias in aging is not universal, and several studies have reported some preference for negative information in healthy older adults [75,76,77]. In our study, the strong memory for emotionally negative destinations can be attributed to the unexpected and perhaps even the surprising effect of encountering emotionally negative destinations. In everyday life, we tend to encounter interlocutors expressing neutral and typically positive emotional expressions, and the exception is to encounter interlocutors experiencing negative emotions. In other words, it is typically rare to relay information to interlocutors experiencing anger or other negative emotions. The high processing of emotionally negative destinations in our control participants may, therefore, be due to the fact that it violates their expectations. In everyday life, positive events, information, and interactions typically occur more frequently than negative ones. Thus, negative events, information, and interactions may be better retained than positive ones because they are the exception and not the norm. This assumption can be supported by the illusory correlation model, which proposes that infrequent events are distinctive and, therefore, draw more attention that frequent events [78]

One limitation of our study is that we only used neutral information. It would be of interest to investigate whether patients with KS would demonstrate diminished memory when relaying negative information. However, and regardless of potential limitations, this study has the merit of answering the open question of the relationship between the decline in memory and decline in emotional processing in KS. KS has been mainly investigated with regard to memory decline, and relatively little research has investigated emotional and affective processing in the syndrome. This is not surprising, as in his original work, Korsakoff [79] did not focus on affective and interpersonal abilities, but rather on memory dysfunction in the syndrome. Therefore, our study does not only contribute to the understating of emotional processing in KS, but also to the inauguration of research on the relationship between the decline in memory and the decline in emotional processing in KS. Our study can also be seen as an attempt to evaluate the relationship between a decline in memory and the decline in social cognition in KS. Research has demonstrated a decline in abilities related to social cognition, such as theory of mind, in KS [25]. This issue is important because theory of mind was found to be related to destination memory; more specifically, research has demonstrated that an intact theory of mind allows deeper processing and, consequently, better recall of our interlocutors [37,80,81]. It therefore follows that a decline in theory of mind, as observed in KS, may contribute to the decline in destination memory in the syndrome.

In summary, our study demonstrates how memory in KS can be influenced by emotion; more specifically, it shows how the emotional attributes of interlocutors may influence destination memory in KS. By doing this, our study contributes to the understating of emotional processing in KS and, more specifically, to the understanding of the relationship between the decline in memory and decline in emotional processing in the syndrome.

## Figures and Tables

**Figure 1 jcm-12-01919-f001:**
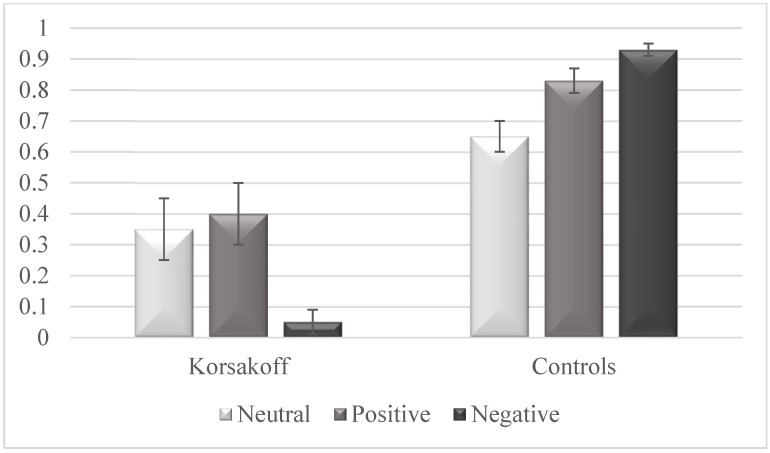
Destination memory performance of patients with Korsakoff’s syndrome and control participants when telling facts to neutral, positive, and negative faces. Scores refer to proportion of hits minus proportion of false recognitions. Error bars represent intervals of 95% within-subjects confidence.

**Table 1 jcm-12-01919-t001:** Cognitive performances of Korsakoff patients and control participants.

	Task	Korsakoff *n* = 25	Controls *n* = 28	Between-Group Comparisons
Episodic memory	Grober and Buschke	6.92 (2.04)	10.50 (1.64)	*t*(51) = 7.07, *p* < 0.001
Working memory	Forward Span	5.56 (1.08)	6.29 (1.30)	*t*(51) = 2.19, *p* < 0.05
	Backward Span	4.00 (0.64)	4.93 (1.01)	*t*(51) = 3.92, *p* < 0.001
Verbal fluency	Verbal Fluency	10.24 (3.98)	16.96 (4.34)	*t*(51) = 5.90, *p* < 0.001

Note. Standard deviations are shown in parentheses; episodic memory score refers to free recall, and maximum score was 16 points; performance in the forward and backward spans refers to the number of correctly repeated digits; performance in verbal fluency refers to number of words beginning with letter P.

## Data Availability

Raw data will be available upon request to the author.

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
