# Peer review of "“What Did I Tell This Sad Person?”: Memory for Emotional Destinations in Korsakoff’s Syndrome"

_jcm, 2023, doi:10.3390/jcm12051919_

Round 1

Reviewer 1 Report

I have some minor comments.

-In the Result section, in Figure 1 scores (Y-axis) refer to proportion of hits
minus proportion of false recognitions. Please, also provide the separate proportions (hits & false recognitions) in the caption of the Figure or in the main text. And discuss intergroup differences, if relevant.

-P.8 L361. Regarding (neutral) fotographs, recognition, and positive/negative information, in experiment 2 of the study of Johnson, et al. (1985) participants with Korsakoff syndrome preferred positive (biographical) information, although they could not remember the information given. Can it be of added value to compare those results with regard to similarities and differences with the current study?

Ref: Johnson MK, Kim JK, Risse G. Do alcoholic Korsakoff's syndrome patients acquire affective reactions? J Exp Psychol Learn Mem Cogn. 1985 Jan;11(1):22-36. doi: 10.1037//0278-7393.11.1.22.

Reviewer 2 Report

HI Marija and JCM office,   your peer review system gives feedback that the form does not work.  I suggest "minor revision".  These are my comments:   1. The patient group is not representative of a normal KS population, because females are far overrepresented. Are there gender differences?
2. The main task is a bit complex to understand. Please try to rephrase the first introduction of the main task in the abstract and introduction section.
3. The findings of the present study seem to contrast the findings of the recently published study by Herrmann et al. (2023) ("the era of our lives: the memory of Korsakoff patients for the first covid-19 pandemic lockdown in the Netherlands" in Consciousness and cognition). Here emotional valence of the patient was related to better memory. Please discusse this possible discrepancy

Small issues:
- KS is an acute -> KS is the chronic phase of the acute Wernicke disease.
- The text below the figure is not in style.
- the reference list is not in style.
- It is unclear why the dsm-iv-tr is used for classification. The DSM 5 is commonly available since 2014.
- Please correct: [Record #5056 is using a reference type undefined in this output style.]   with kind regards,
